# A Dual-Adaptive Equivalent Consumption Minimization Strategy for 4WD Plug-In Hybrid Electric Vehicles

**DOI:** 10.3390/s22166256

**Published:** 2022-08-20

**Authors:** Jianhua Guo, Zhiqi Guo, Liang Chu, Di Zhao, Jincheng Hu, Zhuoran Hou

**Affiliations:** 1State Key Laboratory of Automotive Simulation and Control, Jilin University, Changchun 130022, China; 2Key Laboratory of Bionic Engineering, Ministry of Education, Jilin University, Changchun 130022, China; 3Department of Aeronautical and Automotive Engineering, Loughborough University, Leicestershire LE11 3TU, UK

**Keywords:** energy management strategy (EMS), equivalent consumption minimization strategy (ECMS), four-wheel drive (4WD), plug-in hybrid energy vehicle (PHEV)

## Abstract

Energy management strategies are vitally important to give full play to energy-saving four-wheel-drive plug-in hybrid electric vehicles (4WD PHEV). This paper proposes a novel dual-adaptive equivalent consumption minimization strategy (DA-ECMS) for the complex multi-energy system in the 4WD PHEV. In this strategy, management of the multi-energy system is optimized by introducing the categories of future driving conditions to adjust the equivalent factors and improving the adaptability and economy of driving conditions. Firstly, a self-organizing neural network (SOM) and grey wolf optimizer (GWO) are adopted to classify the driving condition categories and optimize the multi-dimensional equivalent factors offline. Secondly, SOM is adopted to identify driving condition categories and the multi-dimensional equivalent factors are matched. Finally, the DA-ECMS completes the multi-energy optimization management of the front axle multi-energy sources and the electric driving system and releases the energy-saving potential of the 4WD PHEV. Simulation results show that, compared with the rule-based strategy, the economy in the DA-ECMS is improved by 13.31%.

## 1. Introduction

With the increasingly serious energy crisis [1,2,3], plug-in hybrid electric vehicles (PHEVs) have become an effective solution to alleviate the energy problem, with excellent energy-saving characteristics. The unique structure of the multi-energy system endows PHEVs with extraordinary energy-saving potential [4,5]. In PHEVs, the four-wheel-drive PHEV (4WD PHEV) mostly adopts the structure of split-axle four-wheel drive, and the multi-power sources and the multi-power components in the front/rear axles are coupled to form a more complex nonlinear energy system. However, it is a problem to be solved to explore reasonable energy management strategies, optimally manage the multi-energy system and realize the maximum energy-saving potential.

At present, researchers have made remarkable progress in energy management, and the existing energy management strategies are mainly divided into the two following categories [6]: rule-based (RB) energy management strategies [7,8,9] and optimization-based energy management strategies [10,11,12]. Rule-based strategies mainly adopt fixed thresholds and fuzzy logic methods to realize the energy management for the vehicle and the strategies can be equipped with simple structure, mild calculation load and easy online application. However, the RB strategy was developed through expert knowledge and is rather sensitive to driving conditions. In addition, the high-performance continuous control cannot be achieved in actual driving, and the adaptability and robustness of driving conditions are poor. The energy-saving potential of the 4WD PHEV cannot be fully released.

Optimization-based energy management strategies are further divided into global optimization based and instantaneous optimization based [6]. Global optimization-based strategies, such as dynamic programming (DP) [13,14,15] and Pontryagin’s minimum principle (PMP) [16,17,18], can solve the best global control sequence by traversal search under global prior information. However, the global optimization-based strategies are difficult to apply in practice projects because of the heavy computational load. In addition, it is difficult to accurately obtain the future driving condition information in the actual driving, resulting in poor accuracy in the control sequence. The accurate acquisition of prior information and the calculation load in the control sequence have become thorny problems in practice projects. As the 4WD PHEV has a complex energy system composed of multi-power sources and multi-power components, the global-optimization-based strategies fall into the curse of dimensionality for the complex multi-energy system. Existing energy management research for the 4WD PHEV, the global-optimization-based strategies, is only adopted to solve the optimal management of the single energy system [19,20,21]. In addition, prior information of the driving conditions cannot be accurately obtained, resulting in the global optimization strategy unable to give full play to the energy-saving potential of the 4WD PHEV. Instantaneous-optimization-based strategies, including equivalent consumption minimization strategy (ECMS) [21,22,23] and model predictive control (MPC) [24,25], have the adaptability and robustness of driving conditions and the control effect of the vehicle is considered at the same time. The complex nonlinear energy system puts forward higher requirements for the energy management strategy in the 4WD PHEV; therefore, instantaneous-optimization-based strategies have become a better energy management strategy, with good real-time control and economic optimality for the 4WD PHEV.

In the energy system of the 4WD PHEV, energy management is important control technology for multiple energy sources. In order to give full play to the energy-saving potential in the 4WD PHEV, the energy management strategy needs to account for different constraints from the powertrain, driving conditions and the operating states of the vehicle, and weighs the economy and real time of control. As an instantaneous-optimization-based strategy, the MPC strategy takes into account the prior information of future driving conditions and adopts the optimization algorithm to complete the optimal decision-making in the rolling time domain. In simple energy systems, the MPC strategy can realize better energy optimization management and take into account real-time applications and quasi-optimality [26]. However, with an increase in the future prior information dimensions, control step size and control dimensions, the real time and control effect all show a significantly downward trend. In the complex nonlinear energy system of the 4WD PHEV, the coordination output of multiple energy sources increases the control dimension and the calculation load in the MPC, resulting in a control delay and affecting the control effect of the vehicle.

By introducing an equivalent factor, the ECMS equals the consumption of the electric energy to the fuel consumption and solves the local optimal control. According to the current power demand in the vehicle and the operating states of the vehicle components, the optimal energy control sequence is solved at the current time. The ECMS ensures real-time control and economic effect and becomes the research focus of online application. With further research on the ECMS in the energy management strategies of the 4WD PHEV, some experts and scholars have realized the optimal management of energy by introducing an equivalent factor in the energy system of the 4WD PHEV [19,21]. However, in the current energy management strategy research of the 4WD PHEV, the impact of energy management in the power components of the 4WD PHEV on vehicle economy has not been deeply studied and analyzed. The ECMS still needs further exploration to develop the energy-saving potential in the 4WD PHEV. In addition, the value of equivalent factor directly affects the economy of the vehicle as an important parameter of the ECMS. The optimization ability of fixed equivalent factor is inconsistent under different driving conditions and the stable economic optimal control is difficult to guarantee. In the current research, the adaptive equivalent consumption minimization strategy (A-ECMS) is proposed based on the ECMS theory [27,28]. The A-ECMS adopts the historical speed information of the vehicle to predict the future speed information and adjusts the value of equivalent factor to improve the adaptability of driving conditions according to the prior obtained speed information [29,30]. However, the future speed information is variable in time and space, so it is difficult to establish a model to accurately predict the future speed information, even though some methods, such as the convolutional neural network (CNN) [31], the backpropagation neural network (BP-NN) [32] and the Markov chain (MC) [33], have excellent speed prediction effects under some specific driving conditions. However, the accuracy of speed prediction decreases under complex and changeable driving conditions and the speed prediction increases the calculation load, resulting in a poor control effect and poor real-time control of the A-ECMS. Therefore, reducing the unreasonable adjustment in the equivalent factor caused by the inaccuracy of obtaining future prior information is a thorny problem to be solved in the application of the A-ECMS.

In this context, this paper presents a novel energy management strategy for 4WD PHEV, DA-ECMS. The DA-ECMS adopts a multi-layer control architecture to optimize and manage the collaborative output of the multi-energy system. The high level adopts the method of multiple algorithm fusion to complete the classification of driving conditions and the optimization of multi-dimensional equivalent factors offline and completing the identification of driving conditions and the matching of multi-dimensional equivalent factors online. The lower level realizes the optimal management of the multi-energy system in the 4WD PHEV and completing the collaborative control in the power sources’ fuel-electric system (the system composed of engine, generator and battery) and the front/rear axles’ electric driving system. The DA-ECMS comprehensively accounts for the powertrain, driving conditions and the operating states of the vehicle, and balances the economy and real time of control. It gives full play to the energy-saving potential of the 4WD PHEV.

The contributions of this paper are shown as follows:A novel 4WD PHEV energy management strategy, DA-ECMS, is proposed, realizing multi-layer control architecture, combining condition category awareness and the multi-energy system.The collaborative optimization management of the power source fuel-electric system and the front/rear axle electric drive system is completed, giving full play to the energy-saving potential of the 4WD PHEV.The classification of driving conditions and the optimization of multi-dimensional equivalent factors by SOM and GWO are completed offline, and the identification of driving conditions and the matching of multi-dimensional equivalent factors are realized online. The adaptability of the DA-ECMS is improved under different driving conditions.

This study is organized as follows: Section 2 presents an introduction to 4WD PHEV models. Section 3 elaborates the novel methodology of energy management strategy for the 4WD PHEV. Section 4 analyzes and compares simulation results. The conclusions are given in Section 5.

## 2. 4WD PHEV and Model Construction

### 2.1. 4WD PHEV

The 4WD PHEV studied in this paper has a complex energy system with collaborative output in the multi-energy system. The front axle is driven by a mixed structure of different power sources and a single motor is adopted to drive the rear axle. Giving full play to the advantages of the multi-energy collaborative control is the key to realizing energy-saving potential in the 4WD PHEV, and the structure is shown in Figure 1. Under different driving conditions, the collaborative output of multiple energy sources can realize three different operating modes: pure electric mode, series mode and parallel mode, as shown in Table 1. In series mode, the engine drives the generator output energy to drive the vehicle indirectly, but the engine drives directly in parallel mode. Clutch can realize series/parallel mode conversion, changing the form of engine energy output. The detailed parameters of the 4WD PHEV are shown in Table 2.

### 2.2. Mathematic Model of 4WD PHEV

#### 2.2.1. Vehicle Dynamic Model

In the process of establishing the vehicle model, acceleration resistance power, air resistance power, rolling resistance power and slope resistance power become the main factors affecting the dynamic power balance of the vehicle, and the balance equation is shown in Equation (1) [34].
(1)$$P_{req}=\frac{v}{\eta_{t}}(\frac{mgf\cos\alpha_{slap}}{3600}+\frac{C_{D}Av^{2}}{76,140}+\frac{mg\sin\alpha_{slap}}{3600}+\frac{\sigma ma}{3600})$$
where $P_{req}$ is required power of the vehicle; m is the curb weight of the vehicle; g is gravitational acceleration; f is the coefficient of rolling resistance; $\alpha_{slap}$ is the road slope; $C_{D}$ is the air resistance coefficient; A is frontal area of the vehicle; σ is the rotational mass conversion factor; and $\eta_{t}$ is the efficiency of mechanical transmission.

In the vehicle model, multi-power components are input to the half-shaft wheel in the form of torque output. At the wheel, the torque balance equation is shown in Equation (2) [34].
(2)$$T_{req}=(mgf\cos\alpha_{slap}+\frac{C_{D}Av^{2}}{21.15}+mg\sin\alpha_{slap}+\sigma m\frac{dv}{dt})R_{Wheel}$$
where $T_{req}$ is the required torque at the wheel and $R_{Wheel}$ is the wheel rolling radius.

In the driving process of the 4WD PHEV, different modes have different collaborative control strategies of multiple energy sources. Under different modes, the torque relationship of the 4WD PHEV is shown in Equation (3).
(3)$$T_{req}=\{\begin{array}{c} \begin{array}{cc} T_{M\_r}i_{M\_r}\eta_{M\_r}i_{R}\eta_{R}+T_{M\_f}i_{M\_f}\eta_{M\_f}i_{F}\eta_{F} & EV\_Model \end{array}\\ \begin{array}{cc} T_{M\_r}i_{M\_r}\eta_{M\_r}i_{R}\eta_{R}+T_{M\_f}i_{M\_f}\eta_{M\_f}i_{R}\eta_{F} & Serial\_Model \end{array}\\ \begin{array}{cc} (T_{Eng}i_{e}\eta_{e}+T_{M\_f}i_{M\_f}\eta_{M\_f})i_{F}\eta_{F}+T_{M\_r}i_{M\_r}\eta_{M\_r}i_{R}\eta_{R} & Parallel\_Model \end{array} \end{array}$$
where $T_{M\_f}$ is the torque in the front motor; $i_{M\_f}$ is the ratio in the front motor; $\eta_{M\_f}$ is the transmission efficiency in the front motor; $i_{F}$ is final drive ratio for the front axle; $\eta_{F}$ is the efficiency in the front axle final drive; $T_{M\_r}$ is the torque in the rear motor; $i_{M\_r}$ is the ratio for the rear motor; $\eta_{M\_r}$ is the transmission efficiency in the rear motor; $i_{R}$ is final drive ratio for the rear axle; $\eta_{R}$ is the efficiency of the rear axle final drive; $T_{Eng}$ is the torque in the engine; $i_{e}$ is final drive ratio for the engine; and $\eta_{e}$ is the efficiency of the engine.

#### 2.2.2. Engine Model

This paper mainly studies the influence of fuel consumption at a certain speed and torque on the driving economy, but the internal physical characteristics of the engine are not deeply studied. The engine map is established through the relationship among fuel consumption rate, engine speed and torque. The engine map is shown in Figure 2. By looking up the map, the fuel consumption rate of the engine can be quickly and accurately obtained. The instantaneous fuel consumption calculation formula is shown in Equation (4).
(4)$$f_{fuel}(T_{Eng}n_{Eng})=\frac{P_{Eng}(T_{Eng}n_{Eng})\beta_{Eng}(T_{Eng}n_{Eng})t}{3600}$$
where $P_{Eng}$ is the power of the engine and $\beta_{Eng}$ is the fuel consumption rate of the engine.

Under different modes, the operating characteristics in the engine are different. In series mode, the engine speed and torque are completely decoupled from the driving conditions and the operating characteristics are determined by the brake-specific fuel consumption (BSFC). In parallel mode, the engine directly drives the vehicle and the engine speed is in a coupled state with the vehicle speed. The engine speed calculation formula is shown in Equation (5).
(5)$$n_{Eng}=\frac{n_{Wheel}}{i_{F}i_{e}}$$
where $n_{Eng}$ is the speed of engine and $n_{Wheel}$ is the speed of wheel.

#### 2.2.3. Motor/Generator Model

In this paper, permanent magnet synchronous motors (PMSMs) are used in the front/rear motors and generator. The method of experimental calibration is adopted to establish maps of the front/rear motors and generator, and the maps are shown in Figure 3. By looking up the maps, the efficiency of the front/rear motors and generator can be quickly and accurately obtained and the relationship between the consumed power and efficiency is shown in Equations (6)–(8).
(6)$$P_{M\_F}=\{\begin{array}{c} \frac{T_{M\_F}\omega_{M\_F}}{\eta_{M\_F_{Dis}}}\;\;\;\;\;\;\;\;\;\;\;\;\;T_{M\_F}>0\\ T_{M\_F}\omega_{M\_F}\eta_{M\_F_{Cha}}\;\;\;\;\;T_{M\_F}\leq 0 \end{array}$$
(7)$$P_{M\_R}=\{\begin{array}{c} \frac{T_{M\_R}\omega_{M\_R}}{\eta_{M\_R_{Dis}}}\;\;\;\;\;\;\;\;\;\;\;\;\;T_{M\_R}>0\\ T_{M\_R}\omega_{M\_R}\eta_{M\_R_{Cha}}\;\;\;\;\;T_{M\_R}\leq 0 \end{array}$$
(8)$$P_{G}=T_{G}\omega_{G}\eta_{G_{Cha}}$$
where $\omega_{M\_F}$ and $\omega_{M\_R}$ are the speed of the front motor and rear motor, respectively; $\eta_{M\_F_{Dis}}$ and $\eta_{M\_R_{Dis}}$ are the discharge efficiency in the front motor and rear motor, respectively; $\eta_{M\_F_{Cha}}$ and $\eta_{M\_R_{Cha}}$ are the charge efficiency of the front motor and rear motor, respectively; $T_{G}$ is the torque in the generator; $\omega_{G}$ is the speed of the generator; and $\eta_{G_{Cha}}$ is the charge efficiency of the generator.

Because of the mechanical coupling, front/rear motor speed is a fixed proportional relationship with the vehicle speed in the 4WD PHEV, as shown in Equation (9).
(9)$$\{\begin{array}{l} n_{M\_F}=n_{Wheel}i_{F}i_{M\_f}\\ n_{M\_R}=n_{Wheel}i_{R}i_{M\_r} \end{array}$$

#### 2.2.4. Battery Model

As the battery has complex chemical and physical properties, a simple first-order RC battery model is established in this paper, and the battery power relationship is shown in Equation (10) [35]. The method of experimental calibration is adopted to establish the relation diagram for internal resistance, temperature and SOC during charge/discharge. The internal resistance relationship of the battery is shown in Figure 4.
(10)$$P_{b}=V_{OC}I_{b}-I_{b}{}^{2}R_{0}$$
where $P_{b}$ is the power of the battery; $V_{OC}$ is the open-circuit voltage of the battery;$I_{b}$ is the current of the battery; and $R_{0}$ is the internal resistance in the battery.

The terminal current in the battery is obtained from Equation (10), as shown in Equation (11).
(11)$$I_{b}=\frac{V_{OC}-\sqrt{V_{OC}{}^{2}-4P_{b}R_{0}}}{2R_{0}}$$

The change rate in the SOC represents the power consumption degree of the battery as an important parameter, and the relationship of the SOC change rate is shown in Equation (12).
(12)$$S\dot{O}C=-\frac{I_{b}}{Q_{b}}=-\frac{V_{OC}-\sqrt{V_{OC}^{2}-4P_{b}R_{0}}}{2R_{0}Q_{b}}$$
where $Q_{b}$ is the battery capacity.

## 3. Methodology

In this paper, for the complex nonlinear energy system composed of multi-power sources and multi-power components in the 4WD PHEV, a multi-layer energy management control architecture is proposed to combine the prior information of driving conditions and multi-energy optimization management, and gives full play to the energy-saving potential of the 4WD PHEV. The architecture is shown in Figure 5. The high level of control architecture mainly improves the adaptability in the energy management strategy. Self-organizing neural network (SOM) and grey wolf optimizer (GWO) algorithms are adopted to complete the classification of driving conditions and optimization of multi-dimensional equivalent factors offline, realizing the identification of driving conditions and adjustment of equivalent factors online.

The low level of control architecture adopts the DA-ECMS to optimize and manage the collaborative energy output of multiple energy systems in real time. The DA-ECMS realizes the energy distribution of multi-power sources on the front axle of the 4WD PHEV, such as engine, generator and battery. The optimal management between the front/rear motors is completed at the same time. The DA-ECMS balances the economy and real time of control and releases the energy-saving potential of multi-degrees-of-freedom energy system in the 4WD PHEV.

### 3.1. Classification and Online Identification of Driving Conditions Based on SOM

The driving condition characteristics represent the operating state of the vehicle and the future characteristics are accurately and quickly estimated to provide powerful data basis for energy management strategies. There is a strong correlation between road conditions and driving conditions, so this paper mainly adopts the driving speed characteristics of the vehicle to represent the driving conditions and road conditions. At the same time, the driving intention of the driver is also directly reflected by the driving speed.

As an unsupervised learning method, SOM is composed of the input layer and the competition layer. By independently searching for the inherent laws and essential attributes in the samples, SOM adaptively changes the network parameters and structure. A single neuron in the SOM does not play a decisive role and it relies on the synergy of multiple neurons to complete pattern classification. The structure is shown in Figure 6.

In this paper, the characteristics of driving conditions are represented by the average speed, average acceleration, average deceleration and other parameters. The parameters are shown in Table 3. The training set in SOM is constructed from massive dataset of driving condition characteristics to divide into three different categories. In order to provide real-time and accurate driving condition categories for the vehicle energy management strategy, the trained SOM is adopted to identify the category in real time. SOM adopts Euclidean distance to match the best nodes and updates the weight of nodes within the neighborhood, as shown in Equations (13) and (14).
(13)$$d_{j}(x)=\sum_{i=1}^{D}{\left(x_{i}-\omega_{ji}\right)}^{2}$$
(14)$$\omega_{j}(t+1)=\omega_{j}(t)+f_{learn}(t)\times f_{neighbor}(t)\times (x-\omega_{j})$$
where $f_{neighbor}(t)$ is the neighborhood function and $f_{learn}(t)$ is the learning rate.

### 3.2. Optimization of Equivalent Factors Parameters Based on GWO

Based on the principle of grey wolves preying, the grey wolf optimizer (GWO) as a swarm intelligence optimization algorithm was proposed by Mirjalili et al. in 2014 [36]. GWO has the advantages of strong convergence, few parameters, simple structure and easy implementation. The equivalent factors are optimized for different driving conditions in this paper. According to the different operating states and modes in the 4WD PHEV, the values of four equivalent factors in CD-CS series and parallel are optimized, respectively. In the GWO algorithm, the position parameters of the grey wolf represent the equivalent factors in the optimization process and the equivalent fuel consumption is the fitness value of the grey wolf’s current position, so less equivalent fuel consumption means better fitness.

GWO has strict hierarchy of social dominance and it is divided into four classes α,β,δ and ω according to different responsibilities, as shown in Figure 7.

The GWO mainly includes four steps: social hierarchy, tracking, rounding up and hunting, as shown in Figure 8. Firstly, the fitness of each wolf is calculated to construct social hierarchy. Secondly, the three wolves with the best fitness are selected as α,β and δ, respectively, and the remaining wolves are marked as ω. Finally, with the iteration of the wolves, the alpha wolf (α) leads the wolves to gradually approach and hunt the prey. The mathematical expressions of this process are shown in Equation (15).
(15)$$\begin{array}{c} D=CX_{p}(t)-X(t);\\ X(t+1)=X_{p}(t)-AD;\\ A_{x}=2ar_{x1}-a;\\ C_{x}=2r_{x2} \end{array}$$
where t is the number of iterations; $A_{x}$ and $C_{x}$ are the synergy coefficient matrix; $X_{p}$ is the location information of the prey; and $X_{t}$ is the location information of the wolves. In the iterative process, a gradually decreases from 2 to 0; $r_{x1}$ and $r_{x2}$ are random variables in [0, 1].

In the process of hunting, the wolves can identify potential prey through the guidance of α,β and δ. In order to provide wolves with a good ability to identify potential prey, the fitness of grey wolves is updated during each iteration calculation, and the three wolves with the best fitness are selected as α,β and δ respectively. According to the position information of α,β and δ, the position information of the remaining wolves is updated at the next moment. The process expressions are shown in Equations (16) and (17).
(16)$$\left\{ \begin{array}{c} D_{\alpha }=\left|C_{1}X_{\alpha }-X\right|\\ D_{\beta }=\left|C_{2}X_{\beta }-X\right|\\ D_{\delta }=\left|C_{3}X_{\delta }-X\right| \end{array}\right.$$
(17)$$\left\{ \begin{array}{c} X_{1}=X_{\alpha }-A_{1}D_{\alpha }\\ X_{2}=X_{\beta }-A_{2}D_{\beta }\\ X_{3}=X_{\delta }-A_{3}D_{\delta }\\ X(t+1)=\frac{X_{1}+X_{2}+X_{3}}{3} \end{array}\right.$$
where $X_{1}$, $X_{2}$ and $X_{3}$ are the location information of α,β and δ, respectively; X(t+1) is the updated position of the grey wolf; X is the current position of the grey wolf; $D_{\alpha }$, $D_{\beta }$ and $D_{\delta }$ are the distances between the current grey wolf and α,β, δ, respectively; |A| > 1 means that the grey wolves are in scattered mode to search for prey; and |A| < 1 means that the grey wolves focus on hunting for prey.

GWO adopts scattered search for location information of the prey and the prey is locked and captured as the number of iterations t increases. In the process of establishing the GWO model, |A| > 1 means that the wolves are scattered to search for prey and increases the search for the global optimal solution. C, as a random value of [0, 2], has the effect of random weight in the optimization process to avoid falling into the local optimal solution. Finally, the equivalent factors are optimized by GWO under different driving condition categories.

### 3.3. Collaborative Multi-Energy Output Based on DA-ECMS

Multi-power components and multi-energy sources constitute a complex multi-energy system in the 4WD PHEV, and the optimal management of the multi-energy system directly affects the driving economy. In the 4WD PHEV, the front/rear motors directly drive the wheels as the power components and the difference of maps between the front/rear motors, so the rational distribution of energy is the key to improving the economy of the vehicle. In this paper, the DA-ECMS is adopted to complete the optimal management of the multi-energy system and the control architecture is shown in Figure 9.

The DA-ECMS optimizes the distribution of energy output between power sources, while the front/rear axles electric driving system is optimally managed. In order to give full play to the energy-saving potential of the 4WD PHEV, the equivalent factors are adaptively adjusted according to different driving condition categories to improve the driving condition adaptability and robustness and improve the economy of the vehicle. The energy distribution optimization between power sources is shown in Equation (18).
(18)$${\dot{m}}_{eqv}(P_{All}(t),u_{source}(t),t)={\dot{m}}_{f}(P_{All}(t),u_{source}(t),t)+\lambda_{source}{\dot{m}}_{m}(P_{All}(t),u_{source}(t),t)$$
where ${\dot{m}}_{eqv}(P_{All}(t),u_{source}(t),t)$ is the instantaneous equivalent fuel consumption at time *t*; $P_{All}(t)$ is the required power of the vehicle at time *t*; $\lambda_{source}$ is the equivalent factors of the power sources; ${\dot{m}}_{f}(P_{All}(t),u_{source}(t),t)$ is the fuel consumption at time *t*; ${\dot{m}}_{m}(P_{All}(t),u_{source}(t),t)$ is the electric energy consumption at time *t*; and $u_{source}$ is the power source energy output distribution ratio.

The value of $\lambda_{source}$ directly affects the energy distribution between power sources and different driving conditions have different requirements for $\lambda_{source}$. In complex and changeable driving conditions, the single value of $\lambda_{source}$ cannot meet the requirements. According to different driving conditions and modes of the vehicle, the value of $\lambda_{source}$ is adaptively adjusted to improve economy.

Power sources are the energy source of power components and optimize the energy distribution among power components, which can effectively improve the utilization rate. The optimal distribution relation is shown in Equations (19) and (20).
(19)$${\dot{m}}_{All\_Motor}({P^{\prime }}_{All}(t),u_{part}(t),t)=\lambda_{part}\frac{m_{F\_Motor}({P^{\prime }}_{All}(t),u_{part}(t),t)}{\eta_{F\_\mathrm{Motor}}}+\lambda_{part}\frac{m_{R\_Motor}({P^{\prime }}_{All}(t),u_{part}(t),t)}{\eta_{R\_\mathrm{Motor}}}$$
(20)$${P^{\prime }}_{All}(t)=P_{All}(t)\eta_{ele}$$
where ${\dot{m}}_{All\_Motor}({P^{\prime }}_{All}(t),u_{part}(t),t)$ is the instantaneous equivalent fuel consumption of the front/rear motors at time *t*; ${P^{\prime }}_{All}(t)$ is the total power supplied by the power sources to the front/rear motors at time *t*; $\lambda_{part}$ is the equivalent factors of the power components; $m_{F\_Motor}({P^{\prime }}_{All}(t),u_{part}(t),t)$ is the electric energy consumption of the front motor at time *t*; $m_{R\_Motor}({P^{\prime }}_{All}(t),u_{part}(t),t)$ is the electric energy consumption of the rear motor at time *t*; $\eta_{F\_Motor}$ is the efficiency of the front motor; $\eta_{R\_Motor}$ is the efficiency of the rear motor; $\eta_{ele}$ is the transmission efficiency of the electric drive system; and $u_{part}(t)$ is the power component energy output distribution ratio at time *t*.

In order to simplify the non-essential factors in the vehicle dynamics model and reduce the calculated load, only the influence of acceleration resistance power, air resistance power and rolling resistance power on the vehicle are considered in this paper. Equation (1) can be rewritten to obtain $P_{All}$, as shown in Equation (21).
(21)$$P_{All}=P_{a}+P_{f}+P_{w}=\frac{\sigma mv\alpha_{slap}}{1000\eta_{t}}+\frac{mgfv}{1000\eta_{t}}+\frac{C_{D}A}{1632\eta_{t}}v^{3}$$
where $P_{a}$ is the power of the accelerate resistance; $P_{f}$ is the power of the rolling resistance; $P_{w}$ is the power of the air resistance; and v is the vehicle speed (m/s).

Under the requirement of the vehicle, the optimal economy of power sources is realized and the power components in the electric driving system at the front/rear axles are reasonably distributed to improve the electric energy consumption rate at the same time. In this paper, the DA-ECMS adopts the method of multiple distribution to simultaneously complete the collaborative optimal management of Equations (18) and (19), as shown in Equation (22).
(22)$$\begin{array}{l} J(t)=\min\int_{0}^{t}[{\dot{m}}_{eqv}(P_{All}(t),u_{source}(t),\lambda_{source}(t),t)+{\dot{m}}_{All\_Motor}({P^{\prime }}_{All}(t),u_{part}(t),\lambda_{part}(t),t)]dt\\ =\min\int_{0}^{t}\begin{array}{l} {}[\lambda_{part}(t)\frac{m_{F\_Motor}({P^{\prime }}_{All}(t),u_{part}(t),t)}{\eta_{F\_\mathrm{Motor}}}+\lambda_{part}(t)\frac{m_{R\_Motor}({P^{\prime }}_{All}(t),u_{part}(t),t)}{\eta_{R\_\mathrm{Motor}}}\\ +{\dot{m}}_{f}(P_{All}(t),u_{source}(t),t)+\lambda_{source}(t){\dot{m}}_{m}(P_{All}(t),u_{source}(t),t)]dt \end{array}\\ =\min\int_{0}^{t}\begin{array}{l} \left[\lambda_{part}(t)(\frac{m_{F\_Motor}({P^{\prime }}_{All}(t),u_{part}(t),t)}{\eta_{F\_\mathrm{Motor}}}+\frac{m_{R\_Motor}({P^{\prime }}_{All}(t),u_{part}(t),t)}{\eta_{R\_\mathrm{Motor}}}\right)\\ +{\dot{m}}_{f}(P_{All}(t),u_{source}(t),t)+\lambda_{source}(t){\dot{m}}_{m}(P_{All}(t),u_{source}(t),t)]dt \end{array} \end{array}$$

The Hamiltonian function is established to solve the minimum value of the objective function, as shown in Equation (23). In order to solve the optimal control sequence, the general mathematical expression is shown in Equation (24).
(23)$$\begin{array}{cc} H(P_{All}(t),u_{source}(t),u_{part}(t),\lambda_{source}(t),\lambda_{part}(t),t)= & {\dot{m}}_{f}(P_{All}(t),u_{source}(t),t)+\lambda_{source}{\dot{m}}_{m}(P_{All}(t),u_{source}(t),t)+\\ & \lambda_{part}(\frac{m_{F\_Motor}({P^{\prime }}_{All}(t),u_{part}(t),t)}{\eta_{F\_\mathrm{Motor}}}+\frac{m_{R\_Motor}({P^{\prime }}_{All}(t),u_{part}(t),t)}{\eta_{R\_\mathrm{Motor}}}) \end{array}$$
(24)$$\begin{array}{c} H(P_{All}(t),u^{*}{}_{source}(t),u^{*}{}_{part}(t),\lambda_{source}(t),\lambda_{part}(t),t)\leq H(P_{All}(t),u_{source}(t),u_{part}(t),\lambda_{source}(t),\lambda_{part}(t),t)\\ \begin{array}{cc} s.t. & u_{source\_\min}\leq u^{*}{}_{source}(t) \end{array}\leq u_{source\_\max}\\ u_{part\_\min}\leq u^{*}{}_{part}(t)\leq u_{part\_\max}\\ \lambda_{source\_\min}\leq \lambda_{source}(t)\leq \lambda_{source\_\max}\\ \lambda_{part\_\min}\leq \lambda_{part}(t)\leq \lambda_{part\_\max}\\ t_{0}\leq t\leq t_{f} \end{array}$$
where $u^{*}{}_{source}(t)$ is the optimal control sequence of the power sources at time *t*; $u^{*}{}_{part}(t)$ is the optimal control sequence of the power components at time *t*; $u_{source\_\min}$ and $u_{source\_\max}$ are the minimum and maximum distribution ratio of the power source energy output at time t, respectively; $u_{part\_\min}$ and $u_{part\_\max}$ are the minimum and maximum distribution ratio of the power components energy output at time t, respectively; $\lambda_{source\_\min}$ and $\lambda_{source\_\max}$ are the minimum and maximum equivalent factors of the power sources, respectively; and $\lambda_{part\_\min}$ and $\lambda_{part\_\max}$ are the minimum and maximum equivalent factors of the power components, respectively.

In a finite set of Hamilton function, solving the control sequence $u^{*}(u^{*}{}_{source}(t),u^{*}{}_{part}(t))$ makes the Hamilton function obtain the minimum value. The $u^{*}(u^{*}{}_{source}(t),u^{*}{}_{part}(t))$ is the optimal control sequence and the expression is shown in Equation (25).
(25)$$\begin{array}{c} u^{*}(t)=\arg\min H(P_{All}(t),u_{source}(t),u_{part}(t),\lambda_{source}(t),\lambda_{part}(t),t)\\ \begin{array}{cc} s.t. & P_{F\_Motor\_\min}(t)\leq P_{F\_Motor}(t)\leq P_{F\_Motor\_\max}(t) \end{array}\\ P_{R\_Motor\_\min}(t)\leq P_{R\_Motor}(t)\leq P_{R\_Motor\_\max}(t)\\ P_{Engine\_\min}(t)\leq P_{Engine}(t)\leq P_{Engine\_\max}(t)\\ \lambda_{source\_\min}\leq \lambda_{source}(t)\leq \lambda_{source\_\max}\\ \lambda_{part\_\min}\leq \lambda_{part}(t)\leq \lambda_{part\_\max}\\ SOC_{\min}\leq SOC(t)\leq SOC_{\max}\\ t_{0}\leq t\leq t_{f} \end{array}$$
where $P_{F\_Motor\_\min}(t)$ and $P_{F\_Motor\_\max}(t)$ are the minimum and maximum power of the front motor at time t, respectively; $P_{R\_Motor\_\min}(t)$ and $P_{R\_Motor\_\max}(t)$ are the minimum and maximum power of the rear motor at time t, respectively; $P_{Engine\_\min}(t)$ and $P_{Engine\_\max}(t)$ are the minimum and maximum power of the engine at time t, respectively; and $SOC_{\min}$ and $SOC_{\max}$ are the minimum and maximum SOC of the battery, respectively.

The DA-ECMS realizes the optimization management of the multi-energy system and completes the energy management of multi-power sources and multi-power components by solving $u^{*}(u^{*}{}_{source}(t),u^{*}{}_{part}(t))$.The relationship between driving condition categories and optimal equivalent factors are established offline and equivalent factors are adaptively adjusted by identifying the driving condition category online to improve the adaptability and robustness. In complex driving conditions, the DA-ECMS can effectively improve the economy of the 4WD PHEV.

## 4. Simulation Results and Analysis

In this chapter, the equivalent fuel consumption is used as the evaluation standard for the driving economy to compare the six different energy management strategies. Different strategies are shown in Table 4. In this study, due to the 4WD PHEV having multi-power sources and multi-power components, in-depth study of the operating characteristics of different power sources and power components is an important basis for economic analysis. The 4WD PHEV model is established by MATLAB/Simulink (2018a) and the effectiveness of the DA-ECMS is verified under driving conditions of medium–high speed.

### 4.1. Acquisition and Analysis of Future Driving Condition Information

In this paper, SOM identifies the driving condition category in the next 10 s and adaptively adjusts the equivalent factors under different driving modes in the DA-ECMS. As the method of multi-feature judgment is adopted for the driving condition category, the error adjustment for equivalent factors is reduced due to the inaccuracy of prior information.

In the process of driving, the speed information of the past 10 s is adopted to identify the driving condition category of the next 10 s and the category information is updated every 10 s. Due to the short time of updating the driving condition category, the category of the past 10 s has a small error compared to the next 10 s. The simulation results validate that the accuracy rate is about 98.6% and a comparison of the results is shown in Figure 10 and Table 5.

### 4.2. Comparison and Analysis of SOC, Fuel Consumption and Equivalent Fuel Consumption

In this paper, six different energy management strategies are adopted to analyze and compare. The DA-ECMS updates the driving condition category according to the actual information and the equivalent factors are adaptively adjusted to improve the adaptability and robustness by the driving condition category. Compared with the RB strategy, the economy of the DA-ECMS is improved by 13.31% through simulation and comparison. The simulation results are shown in Figure 11, Figure 12 and Figure 13.

Compared with the RB strategy, different energy management strategies have different degrees of economic improvement. The simulation results and comparison data are shown in Table 6.

As shown in Figure 11, the change curves for the SOC are different under different energy management strategies. According to the value of SOC, the 4WD PHEV is divided into CD and CS states. In the CD state, the battery is more inclined to output energy, whereas the engine is more likely to output energy in the CS state. Before 600 s, the SOC of the battery is higher than 0.28, the 4WD PHEV is in CD state and the battery is more inclined to output energy, so the SOC shows a downward trend. In Figure 13, the 4WD PHEV is in pure electric mode for the first 600 s and the battery provides all the energy for the vehicle. The engine is in shutdown state without fuel consumption for the first 600 s. When the SOC is less than 0.28, the vehicle enters CS state and is more inclined to output energy through the engine. Different energy management strategies have different control strategies for the engine, resulting in different driving economies of the vehicle. In Figure 11, the 4WD PHEV enters CS state after 600 s and the SOC shows different trends. Under certain driving conditions, the engine provides extra energy for charging the battery and the SOC shows different upward ranges under different energy management strategies. When the SOC reaches 0.36, the vehicle enters CS state again and is more inclined to output energy from the battery.

The 4WD PHEV has multi-power components in the front/rear motors to drive the vehicle. As the front/rear motors have different maps, the energy optimization management between the front/rear motors also affects the driving economy of the 4WD PHEV. In the first 600 s, the SOC based on the RB and the ECMS strategies decreased more than that based on the H-RB, the D-ECMS, the A-ECMS and the DA-ECMS strategies. As shown in Figure 12, the equivalent fuel consumption based on the RB and the ECMS strategies is higher than other energy management strategies in the first 600 s. For the whole driving cycle shown in Figure 11 and Figure 12, the D-ECMS has different amplitudes in SOC variation and the final value of the equivalent fuel consumption compared with the ECMS. In Figure 12, the final value for the equivalent fuel consumption based on the H-RB and the D-ECMS strategies is significantly lower than that based on the ECMS energy management strategy. Therefore, the reasonable distribution in the front/rear motor energy optimization management can improve the economy of the 4WD PHEV.

In Figure 11, the RB and the H-RB strategies adopt fixed thresholds between the power sources and the range of charging the battery with extra power from the engine is less affected by driving conditions, resulting in a change in the SOC, so it fluctuates greatly. From 600 s to 900 s, the vehicle is in CS state and SOC is always in the rising state. After 900 s, the SOC is greater than 0.36 and the vehicle enters CD state, so the battery provides the main energy and is in a decreasing state. During the whole driving process, the SOC fluctuates between 0.28 and 0.36. In Figure 13, the fuel consumption based on the RB and the H-RB strategies is in step up. Based on the ECMS and the D-ECMS strategies, the energy output is reasonably distributed according to the current operating state and the change in the SOC is affected by driving conditions. In Figure 11, the vehicle is at high speed after 600 s, and the engine provides less power to the charge of the battery for operating in the higher-efficiency region. Therefore, the increase in the SOC is relatively gentle and improves the economy of the vehicle. Due to the different distribution strategies between power components of the ECMS and the DA-ECMS, the change trend for the SOC is different.

The value of equivalent factor has a great impact on the economy of the vehicle control as an important parameter in the equivalent consumption minimization strategy. The ECMS and the D-ECMS strategies have poor adaptability and robustness for adopting fixed initial equivalent factors. Based on the TA-ECMS, the SOM and GWO algorithms are adopted to optimize the initial value of equivalent factors, according to the global characteristics of driving conditions, and the adaptability and robustness of driving conditions are effectively improved. As shown in Figure 11, the vehicle is in the medium-speed stage from 600 s to 1500 s and the SOC increases greatly to provide the power reserve for the high-speed stage in the second half of the driving cycle. In Figure 12, the final value of the equivalent fuel consumption based on the DA-ECMS is obviously lower than the ECMS and the D-ECMS strategies.

Based on the DA-ECMS, the driving condition category is updated every 10 s and the value of the equivalent factors are adaptively adjusted according to different driving condition categories. In Figure 11, the SOC changes in the DA-ECMS show different trends and are more affected by the actual driving conditions. From 600 s to 1300 s, the vehicle is in the medium-speed stage and the engine provides part of the energy. However, the SOC reaches a lower state after 1500 s and the engine gradually becomes the main power output. As the vehicle is at high speed and requires more power, the engine provides less energy for charging the battery, resulting in a slow rise in SOC.

The energy management strategy based on the DA-ECMS can update the value of the equivalent factors in real time according to the actual driving conditions and has excellent adaptability and robustness. As shown in Figure 12, the DA-ECMS has the lowest value of the equivalent fuel consumption under different energy management strategies and the economy of the vehicle is the best. Compared with the RB strategy, the economy of the DA-ECMS is improved by 13.31%.

### 4.3. Qualitative Comparison and Analysis of Engine

In order to analyze and compare the economy of the vehicle under different energy management strategies, analysis of the operating state is essential. This paper mainly focuses on optimizing the operating state of the engine to improve the vehicle economy, so the physical and chemical characteristics of the battery are not studied in depth. Under different control strategies, the engine operating state diagrams are shown in Figure 14 and Figure 15.

As shown in Figure 14, the distribution of the engine operating points is different under six different energy management strategies. The distribution between power sources based on the RB and the H-RB adopts fixed thresholds, so the robustness of driving conditions is relatively poor. As shown in Figure 15, the engine still provides more energy to charge the battery when the power demand of the vehicle is large and the engine has a large torque output. The engine speed at about 2500 r/min has a large torque output, resulting in the engine operating in a lower-efficiency area and increasing the fuel consumption.

According to the current driving condition information, the energy distribution between power sources based on the ECMS and the D-ECMS strategies is optimized. The operating state of the engine is affected by the driving conditions. As shown in Figure 15, the torque output of the engine is lower than the RB and the H-RB, so the driving efficiency area of the engine at 2500 r/min is improved.

According to the driving conditions, the initial value of equivalent factors is reasonably optimized based on the TA-ECMS, improving the adaptability and robustness. As shown in Figure 14 and Figure 15, the engine operating points’ efficiency and stability of torque output are improved based on the TA-ECMS. From 2500 r/min to 3000 r/min of the engine speed, the operating points are obviously improved and most of the points are in the high-efficiency area. According to the real-time driving condition data for the vehicle, the DA-ECMS adaptively adjusts the value of the equivalent factors to further improve the adaptability and robustness of the driving conditions. In Figure 14, compared with the TA-ECMS, the engine speed at about 2000 r/min and 2300 r/min has more operating points distributed in the higher-efficiency area. In Figure 15, the torque output of the engine is obviously adjusted before 2000 s and the equivalent factors are adjusted according to the real-time driving condition information to improve the economy of the whole vehicle. Through the comparison and analysis of the simulation results, the DA-ECMS better adjusts the operating efficiency of the engine and improves the fuel efficiency.

### 4.4. Qualitative Comparison and Analysis of Front/Rear Motors

As the important power components of the 4WD PHEV, the operating state of the front/rear motors has a great impact on the economy of the vehicle. In order to deeply analyze and compare the energy-saving performance of the 4WD PHEV under different energy management strategies, it is necessary to deeply analyze the operating state of power components. The front/rear operating state diagrams are shown in Figure 16, Figure 17, Figure 18 and Figure 19.

As shown in Figure 16 and Figure 17, the H-RB adopts the energy distribution of the front/rear motors. The operating point efficiency for the front motor is obviously improved before the 4000 r/min and the torque output is more stable compared with the RB strategy. Similarly, compared with the ECMS, the operating point efficiency is obviously improved at low speed based on the D-ECMS, the TA-ECMS and the DA-ECMS strategies. Figure 18 and Figure 19 show the operating state of the rear motor and the state is obviously adjusted under different control strategies. Compared with the RB and the ECMS strategies, the operating point efficiency distribution of the rear motor is obviously improved from 1500 to 7000 r/min based on the H-RB, the D-ECMS, the TA-ECMS and the DA-ECMS. In the 4WD PHEV studied in this paper, the maps of the front/rear motors are different and the overall efficiency of the rear motor is higher than the front motor. Therefore, in order to minimize the overall energy consumption in the front/rear motors, the operating points of the rear motor are greatly adjusted to operate in the higher-efficiency area and the economic improvement effect is more significant. According to the above simulation results, the optimized management of the front/rear motors can effectively improve the power efficiency of the 4WD PHEV electric driving system and improve the economy of the vehicle.

## 5. Conclusions

In this paper, a multi-layer energy management control architecture is proposed to optimize management of the 4WD PHEV multi-energy system, combining the driving condition categories with the DA-ECMS to give full play to the energy-saving potential of the 4WD PHEV and improve the adaptability of driving conditions. The high level adopts SOM and GWO to classify the driving condition categories and optimize the multi-dimensional equivalent factors offline, realizing the identification of the categories by SOM and matching the equivalent factors online. The low level adopts the DA-ECMS to distribute the multi-degrees-of-freedom energy output between the 4WD PHEV multi-power sources and multi-power components and balancing the real time and economic performance of the control. The simulation results are compared by MATLAB/Simulink in this paper and the economy of the DA-ECMS is improved by 13.31% compared with the RB strategy.

However, the errors in the predicted condition categories cannot accurately reflect the future driving condition information and the final control effect cannot reach the best state. In future scientific research, the predicted condition category errors shall be solved to further improve the economy of the 4WD PHEV.

## Figures and Tables

**Figure 1 sensors-22-06256-f001:**
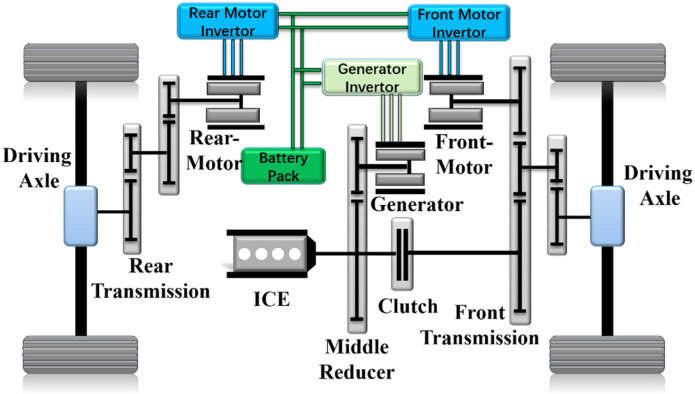
The schematic of the 4DW PHEV configuration.

**Figure 2 sensors-22-06256-f002:**
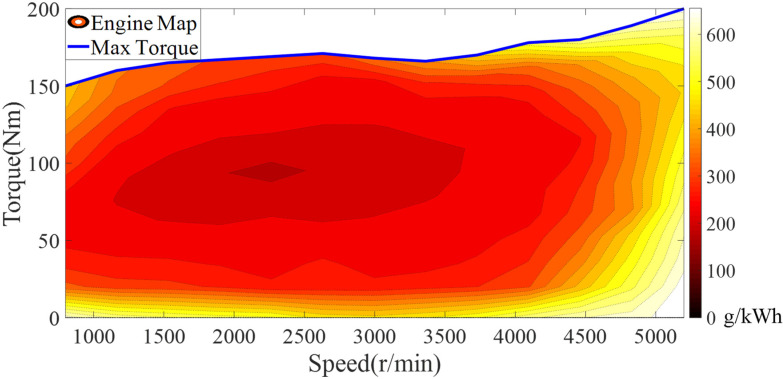
The fuel consumption map of the engine.

**Figure 3 sensors-22-06256-f003:**
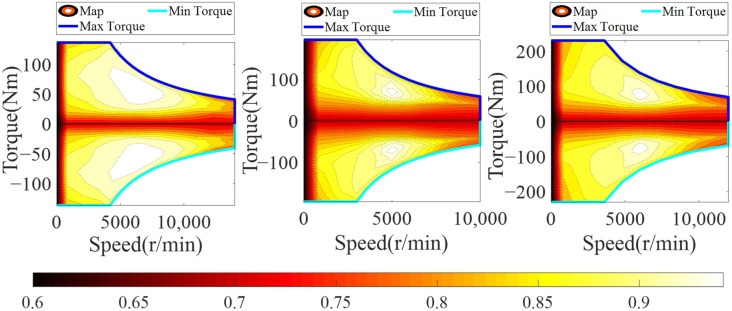
Efficiency maps of front/rear motors and generator.

**Figure 4 sensors-22-06256-f004:**
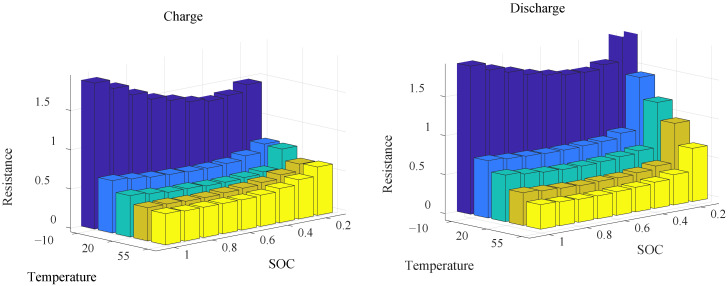
The battery of discharging/charging internal resistance diagrams.

**Figure 5 sensors-22-06256-f005:**
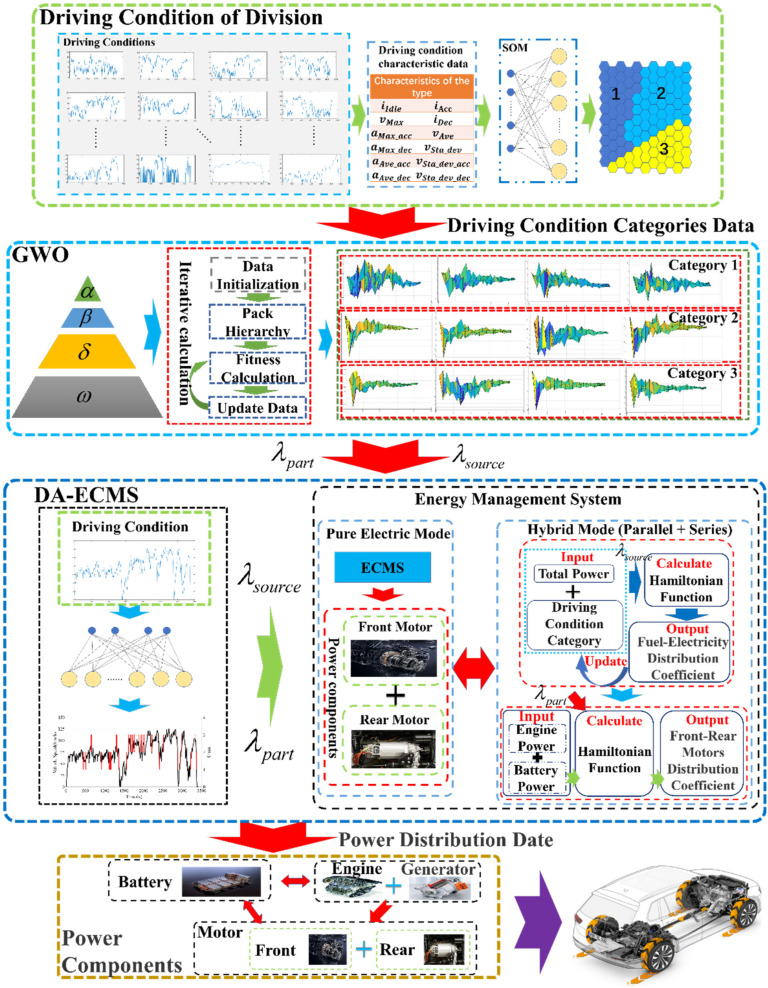
Illustration of the energy management strategy process.

**Figure 6 sensors-22-06256-f006:**
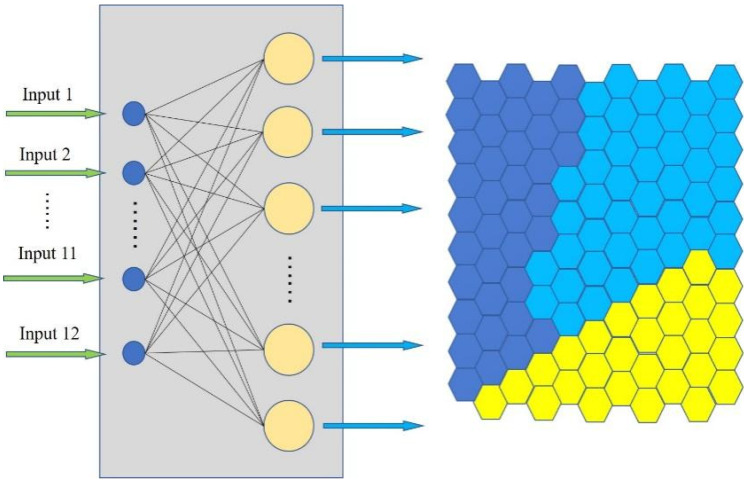
The architecture of the SOM.

**Figure 7 sensors-22-06256-f007:**
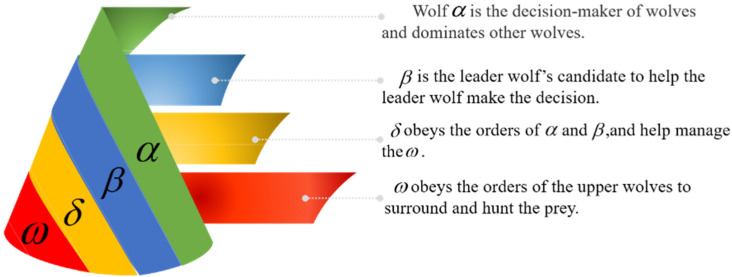
The social dominance hierarchy diagram of GWO.

**Figure 8 sensors-22-06256-f008:**
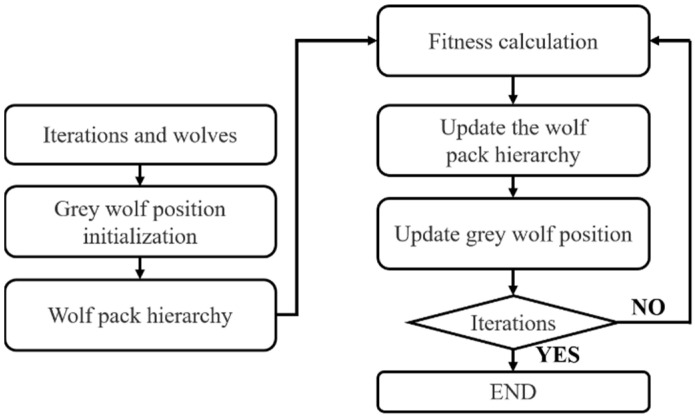
The architecture of the GWO.

**Figure 9 sensors-22-06256-f009:**
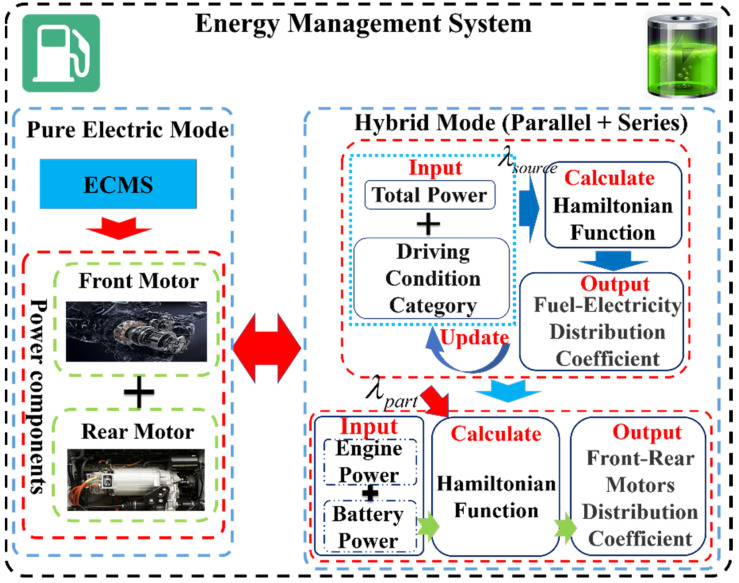
The architecture of the DA-ECMS.

**Figure 10 sensors-22-06256-f010:**
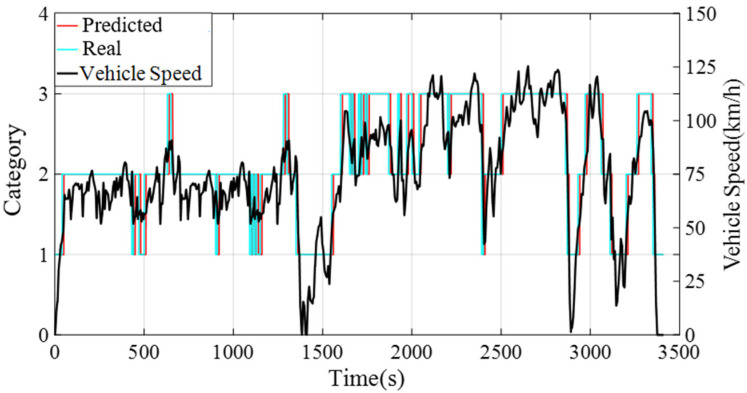
The error comparison of the past 10 s and the next 10 s.

**Figure 11 sensors-22-06256-f011:**
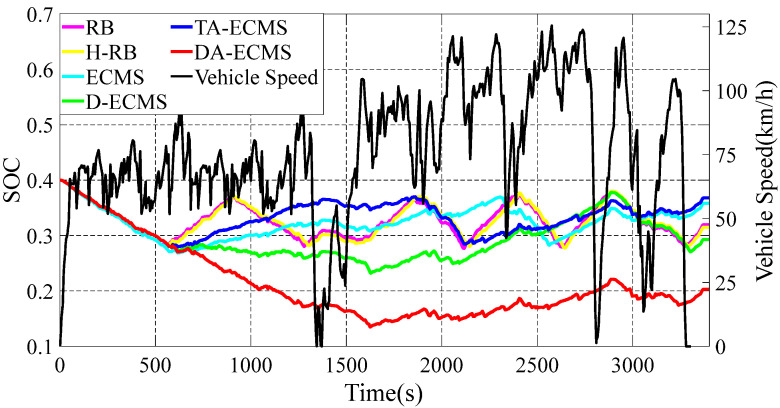
Variation curves for the vehicle speed and the SOC under different strategies.

**Figure 12 sensors-22-06256-f012:**
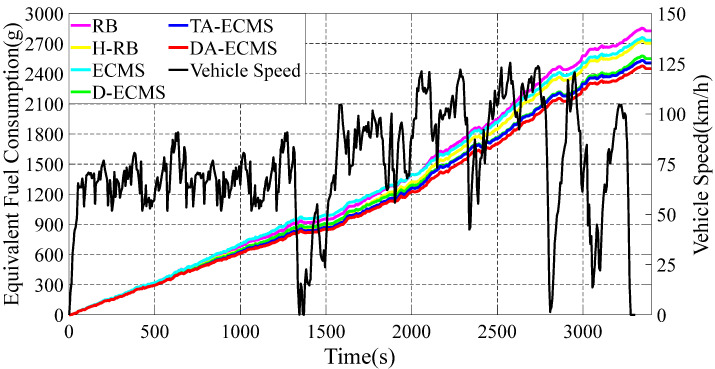
Variation curves for the vehicle speed and the equivalent fuel consumption under different strategies.

**Figure 13 sensors-22-06256-f013:**
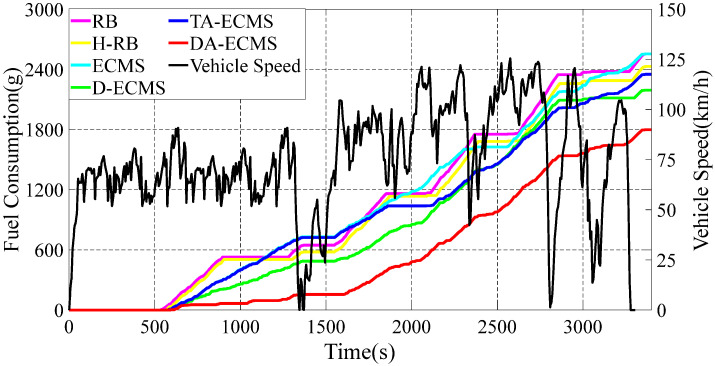
Variation curves for the vehicle speed and the fuel consumption under different strategies.

**Figure 14 sensors-22-06256-f014:**
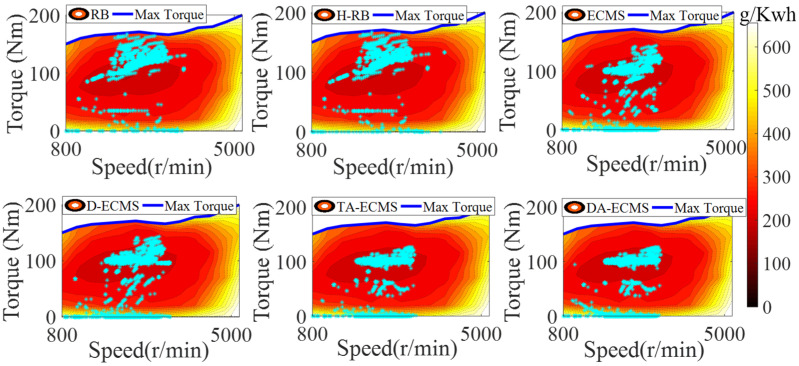
Engine operating points of energy management strategies.

**Figure 15 sensors-22-06256-f015:**
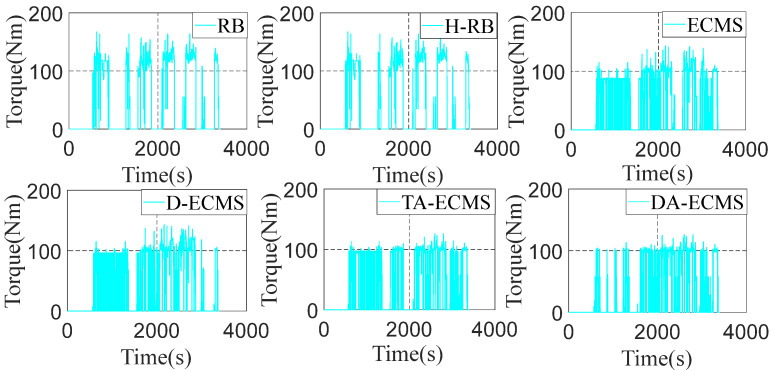
Variation curves for engine torque under different strategies.

**Figure 16 sensors-22-06256-f016:**
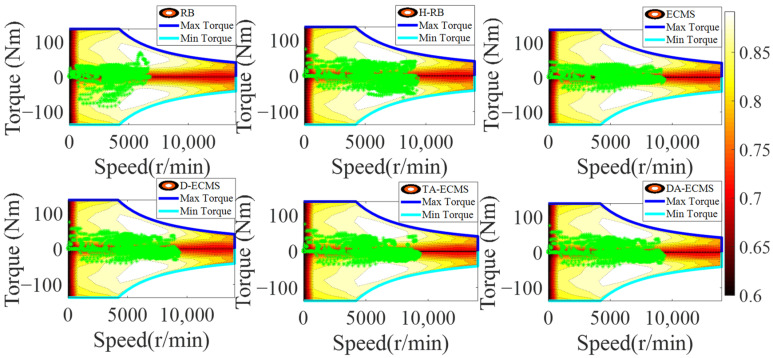
Front motor operating points of energy management strategies.

**Figure 17 sensors-22-06256-f017:**
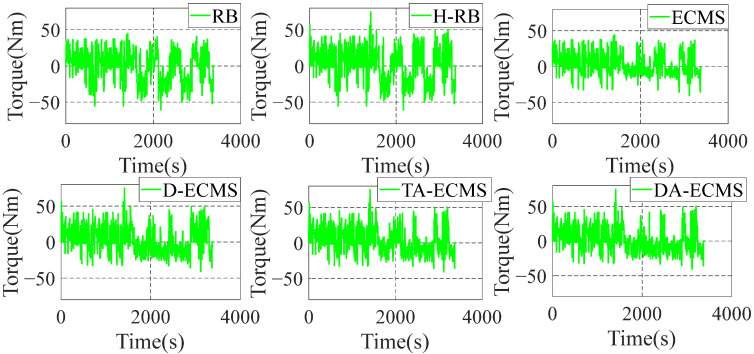
Front motor torque of energy management strategies.

**Figure 18 sensors-22-06256-f018:**
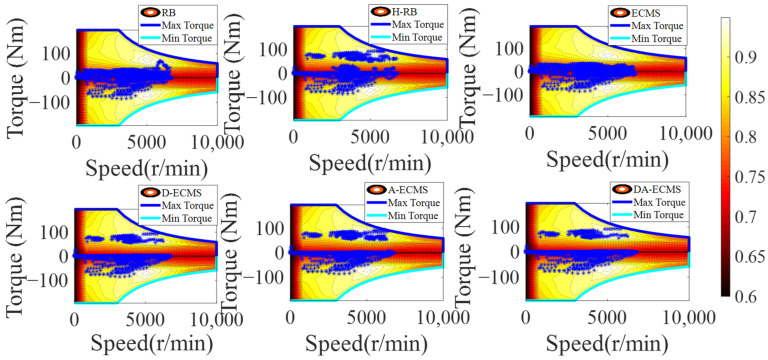
Rear motor operating points of energy management strategies.

**Figure 19 sensors-22-06256-f019:**
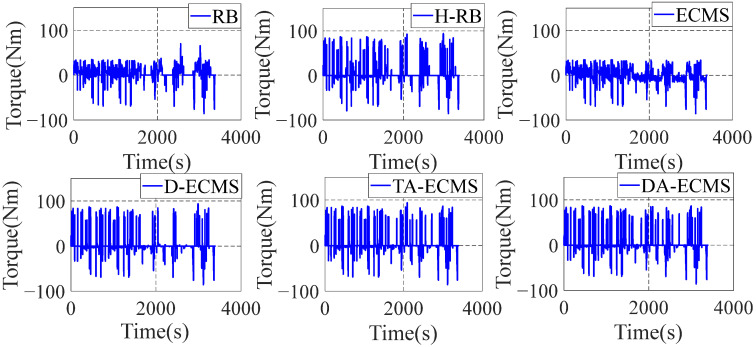
Rear motor torque of energy management strategies.

**Table 1 sensors-22-06256-t001:** Operating states of vehicle components under different modes.

Operating Modes	Illustration
Pure electric mode	The battery provides all the power for the front/rear motors to drive the vehicle, and the engine and generator are in shutdown state.
Series mode	The engine drives the generator to provide electric energy for the front/rear motors, and the battery also provides electric energy output.
Parallel mode	The clutch is closed, and the engine directly drives the vehicle. The front/rear motors assist the engine to drive the vehicle.

**Table 2 sensors-22-06256-t002:** Vehicle and dynamic components parameters in the 4DW PHEV.

Parameter	Unit	Value
Vehicle Mass	kg	1860
Vehicle Maximum velocity	km/h	170
Wheel rolling radius	m	0.35
Frontal area	m^2^	2
Engine maximum power	kW @ rpm	110 @ 5200
Engine maximum torque	Nm @ rpm	200 @ 5200
Front motor maximum power	kW	60
Front motor maximum torque	Nm	137
Rear motor maximum power	kW	61
Rear motor maximum torque	Nm	195
Battery capacity	kWh	15
Battery rated voltage	V	300

**Table 3 sensors-22-06256-t003:** The driving condition characteristics table.

Driving Condition Category Characteristics	Unit	Symbol
Idle time/Total time	%	$i_{Idle}$
Maximum speed	m/s	$v_{Max}$
Maximum acceleration	m/s^2^	$a_{Max\_acc}$
Maximum deceleration	m/s^2^	$a_{Max\_dec}$
Average acceleration	m/s^2^	$a_{Ave\_acc}$
Average deceleration	m/s^2^	$a_{Ave\_dec}$
Acceleration time/Total time	%	$i_{Acc}$
Deceleration time/Total time	%	$i_{Dec}$
Average speed (Excluding parking time)	m/s	$v_{Ave}$
Standard deviation of speed	m/s	$v_{Sta\_dev}$
Standard deviation of acceleration	m/s^2^	$a_{Sta\_dev\_acc}$
Standard deviation of deceleration	m/s^2^	$a_{Sta\_dev\_dec}$

**Table 4 sensors-22-06256-t004:** Table of different control strategies.

Energy Management Strategy	Illustration
RB	The RB strategy is adopted to optimize the energy management of the power sources, and power components adopt fixed energy distribution ratio.
H-RB	The RB strategy is adopted to optimize the energy management of the power sources, and power components adopt ECMS to optimize the energy management.
ECMS	The ECMS is adopted to optimize the energy management of the power sources, and power components adopt fixed energy distribution ratio.
D-ECMS	Both power sources and power components adopt the ECMS to optimize the energy management.
TA-ECMS	Under the method of total optimizing the initial value of equivalent factors, both power sources and power components adopt the ECMS to optimize the energy management.
DA-ECMS	Under the method of instantaneous optimizing the initial value of equivalent factors, both power sources and power components adopt the ECMS to optimize the energy management.

**Table 5 sensors-22-06256-t005:** The predicted category and real category comparison table.

Total Sampling Time (s)	Same Category Time (s)	Different Category Time (s)	Accuracy Rate
3410	3362	48	98.6%

**Table 6 sensors-22-06256-t006:** Comparison of the simulation results and the data under different energy management strategies.

Control Strategy	Terminal SOC	Equivalent Fuel Consumption (g)	Economy (Relative to RB)
RB	0.320	2826	
H-RB	0.315	2703	4.35%
ECMS	0.358	2733	3.29%
D-ECMS	0.293	2549	9.80%
TA-ECMS	0.368	2506	11.32%
DA-ECMS	0.203	2450	13.31%

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
