# Peer review of "A Dual-Adaptive Equivalent Consumption Minimization Strategy for 4WD Plug-In Hybrid Electric Vehicles"

_sensors, 2022, doi:10.3390/s22166256_

Round 1

Reviewer 1 Report

It's understandable that this paper does not focus on the internal physical characteristics of the ICE. But the engine map shown in Fig.2. starts with 0rpm. Even if use of the ICE by 0-800 rpm is somehow prevented in the simulation model this could course misunderstandings.

Same goes for the battery diagrams at Fig.4. There is no explication wether the diagram shows a tecnical SOC or a users SOC. Tecnical SOC can't go down to 0. Again, even if it is somehow prevented in the simulation model this could course misunderstandings.

In line 282/ equation (14) I may have found a little tipo: "f_learn(t) is the neighborhood function"

It would have been interesting to see the change of different driving modes caused by different strategies: Diagram with EV-, Series- and parallel-Mode over time and strategie.

Are there intentions to use the DA-ECMS for power-split hybrid, too? 

Author Response

Thank you very much for your suggestions. According to your suggestions, the following modifications have been made.

  1. The operating characteristic diagrams of the engine has been corrected, and the operating range of 0-800r/min is avoided. The diagrams are shown in Fig.2 and Fig.14.
  2. The internal resistance characteristic diagram of the battery has been corrected, and the state of SOC reaching 0 has been avoided. The internal resistance characteristic diagram shown in Fig.4.
  3. In equation (14), the f_learn(t) has been corrected.
  4. This article establishes the vehicle model by MATLAB/Simulink, the driving modes are mainly determined by the vehicle power, SOC and vehicle speed. The diagrams with EV- mode, Series-mode and parallel-mode over time and strategie become messy and unclear, so the diagrams are not shown in the article. However, table 1 is added to illustrate the operating status of vehicle components under the different driving modes.
  5. In the future research, I will try to realize the DA-ECMS for power-split hybrid, and this article mainly realizes the application of the DA-ECMS in Fig.1 configuration. Thank you very much for your suggestions.

Reviewer 2 Report

The paper aims to minimize fuel consumption via a novel dual-adaptive minimization strategy for the complex multi-energy system in the 4WD PHEV.

The paper is well-written, clear, and presents an exciting approach.

However, before publication, significant updates are required:

1) fix typos and improve English style;

2) Insert references for equations. For instance, for equation 10 see ([Maia, R., Silva, M., Araújo, R., & Nunes, U. (2015). Electrical vehicle modeling: A fuzzy logic model for regenerative braking. Expert systems with applications42(22), 8504-8519.]). For equation 1 and vehicle dynamics, see ([Caiazzo, B., Coppola, A., Petrillo, A., & Santini, S. (2021). Distributed nonlinear model predictive control for connected autonomous electric vehicles platoon with distance-dependent air drag formulation. Energies14(16), 5122.]).

3) Authors should add a comparison with a more efficient control strategy (exploiting, for instance, model predictive control). For instance, see ([Xu, W., Chen, H., Zhao, H., & Ren, B. (2019). Torque optimization control for electric vehicles with four in-wheel motors equipped with regenerative braking system. Mechatronics57, 95-108.])

Author Response

Thank you very much for your suggestions. According to your suggestions, the following modifications have been made.

  1. In this article, English errors and English style have been corrected.
  2. According to your suggestion, the formula reference has been inserted into the article, as shown in equation(1) and equation(10).
  3. This paper has made a simple description and comparison of the MPC. Such as line 76-86. Because this paper mainly focuses on the comparison and innovation of different ECMS. There is no further analysis and comparison of the MPC simulation results in Section 4.

Reviewer 3 Report

Dear Authors,

This paper presented an energy minimization strategy for 4WD Plug-in Electric Vehicles. Following are my comments:

The concept has been well presented and the methodology is supported by simulation results. Minor changes/additions must be made to further improve the paper.

Please check the paper thoroughly for spelling errors, grammatical mistakes and sentence formation errors. On page 10, 'grey' is misspelled as 'gray'. On page 19, 'better' is misspelled as 'batter'.

Move the title of table 5 to the next page.

Kindly explain the advantages of SOM for the classification of driving conditions.

Since the driving conditions also depend on the road conditions, can they be incorporated into the proposed energy management strategy?

Please explain how the control strategy is implemented inside the Plug-in Electric Vehicle.

Should the vehicle driver provide any inputs to implement the proposed energy management strategy?

Kindly mention the version of Matlab/Simulink that has been used to perform the simulations.

Please show the equivalent circuit of the battery to justify equation 10.

What are the final driving conditioning categories identified by the energy management strategy? Kindly present the information in a tabular format.

Author Response

Thank you very much for your suggestions. According to your suggestions, the following modifications have been made.

  1. In this article, English errors have been corrected.
  2. The title format of the table has been corrected.
  3. In this article, the advantages of SOM classification have been explained, such as line 272-276.
  4. In this article, the driving conditions, road conditions and driver's driving intention have been explained, such as line 266-271.
  5. The control strategy implemented in the plug-in electric vehicle is shown in Figure 5, which illustrates the control logic and implementation forms of different components under the DA-ECMS control strategy. Figure 9, table 1 and table 4 also illustrate the method of implementing control.
  6. This article uses the 2018a version of the MATLAB software, which has been corrected in the article, such as line 429.
  7. Equation(10) is derived from the first-order RC model, and the Equation(10) is given with reference to [37].
  8. This article uses the characteristics of driving conditions to classify driving, as shown in Table 3. The actual driving information of the vehicle is adopted to identify the driving category. Such as line 432-443. Selecting equivalent factors according to different driving categories, as shown in Figure 5. In Section 3.2, it is explained how different categories and equivalent factors establish correlation. This driving category method is only applied to DA-ECMS and TA-ECMS. However, the driving categories of DA-ECMS and TA-ECMS are also different, as shown in the table 4.

Round 2

Reviewer 2 Report

Authors worked a lot to improve the article.